# Learning is enhanced by tailoring instruction to individual genetic differences

**David G Mets[1,2]\*, Michael S Brainard[1,2,3,4]\***

[1]Center for Integrative Neuroscience, University of California, San Francisco, San Francisco, United States; [2]Howard Hughes Medical Institute, University of California, San Francisco, San Francisco, United States; [3]Department of Physiology, University of California, San Francisco, San Francisco, United States; [4]Department of Psychiatry, University of California, San Francisco, San Francisco, United States

**Abstract** It is widely argued that personalized instruction based on individual differences in learning styles or genetic predispositions could improve learning outcomes. However, this proposition has resisted clear demonstration in human studies, where it is difficult to control experience and quantify outcomes. Here, we take advantage of the tractable nature of vocal learning in songbirds (*Lonchura striata domestica*) to test the idea that matching instruction to individual genetic predispositions can enhance learning. We use both cross-fostering and computerized instruction with synthetic songs to demonstrate that matching the tutor song to individual predispositions can improve learning across genetic backgrounds. Moreover, we find that optimizing instruction in this fashion can equalize learning differences across individuals that might otherwise be construed as genetically determined. Our results demonstrate potent, synergistic interactions between experience and genetics in shaping song, and indicate the likely importance of such interactions for other complex learned behaviors.

DOI: https://doi.org/10.7554/eLife.47216.001

**\*For correspondence:**
dmets@phy.ucsf.edu (DGM);
msb@phy.ucsf.edu (MSB)

**Competing interests:** The authors declare that no competing interests exist.

## Introduction

Recent studies in human populations have demonstrated strong genetic influences on academic achievement (*Branigan et al., 2013*; *Lee et al., 2018*; *Okbay et al., 2016*; *Rietveld et al., 2013*). This raises the question of whether genes place immutable bounds on achievement or whether experiential factors could amplify or dampen the impact of genetic variation. A particularly intriguing possibility is that customizing instruction based on individual differences in learning styles or genetic predispositions could improve learning outcomes (*Asbury and Plomin, 2013*; *Moser and Zumbach, 2018*; *Pashler et al., 2009*; *Plomin, 2014*). However, despite widespread interest in the potential value of 'personalized' instruction, it has been difficult to evaluate this idea in human populations, where it is challenging to control genetic variation, manipulate experience and quantify outcomes.

Birdsong affords an attractive system for studying how tailoring instruction based on genetic differences influences learning outcomes. Song, like human speech, is a complex vocal behavior that is learned during early life from adult vocal models (*Doupe and Kuhl, 1999*). Young birds listen to a 'tutor song' (usually that of their father), and through practice develop vocalizations that closely match this target (e.g. *Figure 1A*). Although much prior work has focused on how experience shapes song learning, genetic predispositions also contribute to learned song structure at both the species and individual levels (*Fehér et al., 2009*; *Gardner et al., 2005*; *Marler and Peters, 1977*; *Marler and Peters, 1982*; *Marler and Peters, 1988*; *Mets and Brainard, 2018a*; *Mundinger, 1995*; *Mundinger and Lahti, 2014*; *Podos et al., 1999*; *Soha and Marler, 2000*; *Thorpe, 1954*). For

**eLife digest** Some people do better at school than others, and some of this difference comes down to genes. But do genes place fixed limits on an individual's academic potential? Or is it possible to increase or decrease the impact of genes by changing how a person is taught? One possibility is that individuals learn best in different ways, and that tailoring instruction to suit individual learning styles could improve learning outcomes. But despite widespread interest in this idea, testing it systematically has proven challenging.

Mets and Brainard have therefore taken a new approach by testing the idea in a songbird called the Bengalese finch. Birdsong is a complex behavior learned in a similar way to human speech. Young birds listen to a tutor song – usually that of their father – and learn to mimic it through trial and error. But some songbirds learn better than others. By swapping eggs between nests, Mets and Brainard show that genetic offspring often learn the father's song more accurately than birds fostered in from another nest. This might be because the father and offspring share genetic characteristics that contribute to the sound of the father's song. Birds with the same genes will thus find it easier to learn the same song.

Alternatively, it could be that father birds spend more time teaching their genetic offspring than young they have fostered. To control for this possibility, Mets and Brainard played computer-generated songs to juvenile birds from different nests that had all been raised by non-singing females. Some of the songs had a fast tempo, others were slow, and a third set were in between. The results showed that juveniles learned most successfully when the training song had a similar tempo to their father's song. This was true even though none of the birds had ever heard their father sing.

The findings thus suggest that tailoring instruction to suit an individual's natural learning tendencies – which depend on their genes – can enhance learning. Without knowing about this effect, it would be easy to assume that some of the songbirds in the current study were simply poorer learners than others. But in fact, optimizing instruction for each individual's genetic background reduced the differences between individuals. If learning in humans is similar to vocal learning in birds, there could be broad implications for education.

DOI: https://doi.org/10.7554/eLife.47216.002

example, while many species can learn aspects of heterospecific song, birds will preferentially acquire species typical song structure (spectral content and ordering of syllables) when tutored with a combination of heterospecifc and conspecific songs (*Marler and Peters, 1977*; *Marler and Peters, 1982*; *Marler and Peters, 1988*; *Mundinger, 1995*; *Mundinger and Lahti, 2014*; *Podos et al., 1999*; *Soha and Marler, 2000*; *Thorpe, 1958*). Consistent with this, deviations between a tutor song and species typical song can lead to poor copying of the tutor song (*Lahti et al., 2011*; *Marler and Peters, 1988*; *Podos, 1997*; *Podos et al., 2004*). Moreover, we have previously found that even within a single-species colony of Bengalese finches (*Lonchura striata domestica*), there is a strong heritable predisposition for individuals to produce songs at differing tempos (*Mets and Brainard, 2018a*). The presence of such heritable biases for learned song structure, together with the ease of controlling instructive experience and quantifying learning outcomes, renders song learning in Bengalese finches particularly suitable for testing whether tailoring instruction in accordance with individual genetic predispositions can enhance learning.

## Results

Within our genetically heterogeneous Bengalese finch colony, we found that there was a broad range in the quality of song learning. Many juveniles that were reared conventionally in their home nests and tutored by their genetic fathers learned to copy tutor song flexibly and with high fidelity (e.g. *Figure 1Ai–ii*, tutor vs. learned song). However, across individuals, learning could range from nearly perfect (e.g. *Figure 1B*, tutor song vs. tutee song, top) to extremely poor (e.g. *Figure 1B*, tutor song vs. tutee song, bottom; compare with isolate song). We quantified song learning using the *Song Divergence (SD)*, a measure that estimates how much of the spectral content of syllables in the tutor song is absent from the learned song (*Mets and Brainard, 2018b*); hence, an SD of 0

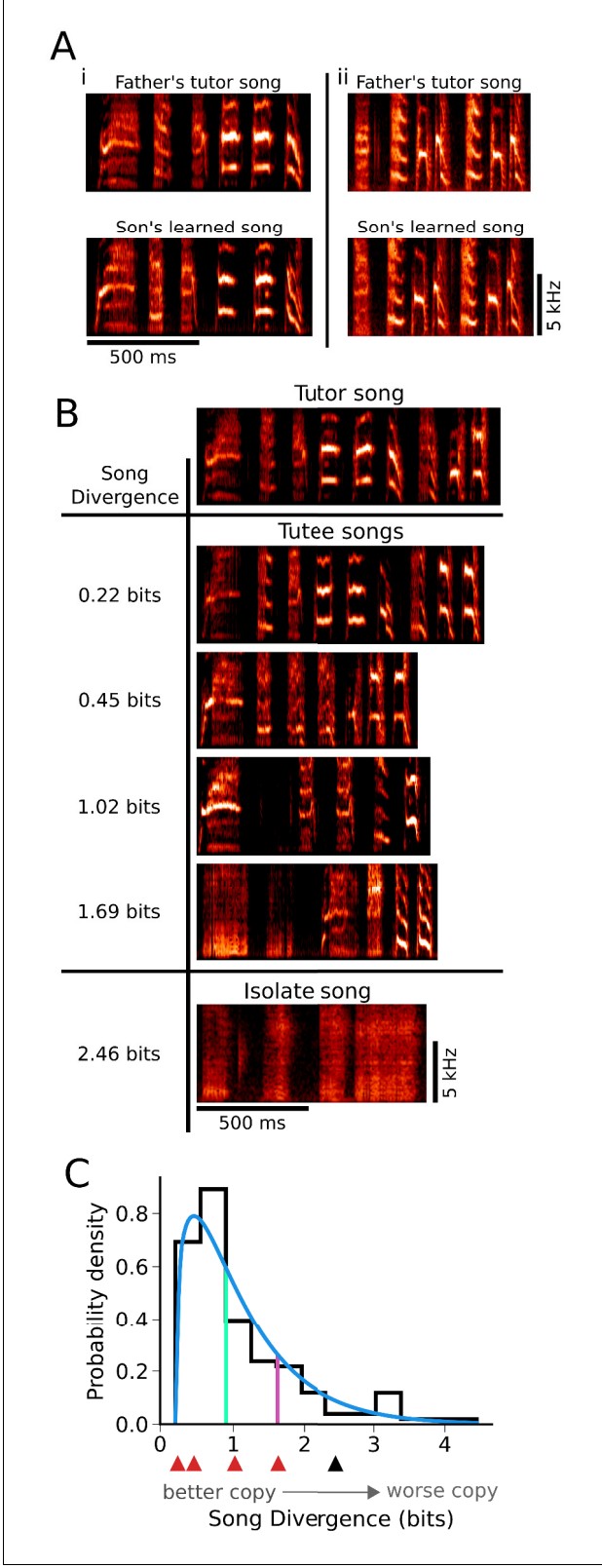

**Figure 1.** There is broad variation in the quality of learning across individuals. (**A**) Spectrograms illustrating learning of song from different tutors. Adult song is composed of a series of discrete units of sound (syllables) separated by silence, with each syllable containing characteristic spectral content. Two fathers' songs (top: i and ii) had distinct syllable structure that was well learned by their offspring (bottom: i and ii). (**B**) Spectrograms of songs

*Figure 1 continued on next page*

*Figure 1 continued*

exemplifying the range of variation in learning present under conventional rearing and tutoring conditions. Similarity between the learned songs of individuals (Tutee songs) and the song of their tutor (Tutor song) is indicated by Song Divergence (SD) scores at left. SD indicates spectral content from a tutor song that is missing from the learned song, such that lower SD scores indicate better learning (see Materials and methods). The song of a bird that had no exposure to a tutor (isolate song) is shown for comparison. (C) The distribution of learning outcomes for 142 birds that were reared and tutored in their home nests. A gamma distribution fit to the data (blue line) and the 50th percentile (green) and 80th percentile (red) indicate that learning outcomes were skewed towards good learning (smaller SD values). Red and black arrows correspond to the learned 'tutee' songs and 'isolate' song presented in panel B.

DOI: https://doi.org/10.7554/eLife.47216.003

indicates a song that perfectly matches the tutor song, while increasing values of SD (quantified in bits) indicate songs that are progressively worse copies of the tutor. The SD is computed in an automated fashion from a large set of syllables randomly selected from each bird's song, and provides a measure that corresponds well with human assessments of song learning across a broad range of learning quality (see Materials and methods) (*Mets and Brainard, 2018b*). Prior work indicates that variation in the quality of song learning (*Figure 1C*) could be influenced by both experiential and genetic factors (*Chen et al., 2016*; *Doupe and Kuhl, 1999*; *Fehér et al., 2009*; *Gardner et al., 2005*; *Marler and Peters, 1988*; *Marler and Peters, 1982*; *Marler and Peters, 1977*; *Mets and Brainard, 2018a*; *Mundinger, 1995*; *Mundinger and Lahti, 2014*; *Podos et al., 1999*; *Soha and Marler, 2000*; *Tchernichovski et al., 1999*; *Thorpe, 1958*). Here, we were interested in the possibility that a component of this variation could be explained by an interaction between these factors - specifically, whether matching instructive experience to genetic predispositions of individuals could improve learning outcomes.

To investigate whether alignment between tutoring experience and individual genetic predispositions influences learning outcomes, we compared how well a tutor's song was learned by his own genetic offspring relative to how well it was learned by birds that were cross-fostered from other nests (*Figure 2A*). We reasoned that many of the genetic factors that contribute to the structure of a father's song would also be passed on to his offspring. Hence, we expected that a tutor's song would be better matched to the genetic predispositions of his own offspring than to those of cross-fostered birds (*Plomin et al., 1977*). We hypothesized that if the correspondence between tutor song instruction and individual genetic biases influences learning, then home-reared birds would learn better than cross-fostered birds. Eight breeding pairs served as parents and provided tutoring to both their own home-reared progeny and to cross-fostered birds (see Materials and methods). To ensure that cross-fostered birds were not exposed to the songs of their genetic fathers, we transferred eggs to the nests of foster parents within 36 hr of laying, prior to the development of the peripheral auditory system (*Murray et al., 2013*; *Yamasaki and Tonosaki, 1988*). For both cohorts, juveniles were reared with their tutors from hatching until adulthood (~120 days of age), at which point their songs were recorded for analysis. In order to assess any learning differences in the context of the same tutor song and parental environment, we conducted paired comparisons of the median quality of learning within each nest for home-reared versus fostered juveniles.

Across nests, genetic offspring (home-reared birds) learned the tutor song significantly better than cross-fostered birds (*Figure 2B,C*; $p < 0.005$, Wilcoxon signed-rank test). This suggests that better alignment between the acoustics of the tutors' songs and the individual genetic predispositions of their offspring improved learning outcomes for home-reared birds. However, learning in these experiments also could have been facilitated for home-reared birds by genetic contributions to other aspects of instructive experience, such as the amount or quality of tutoring directed at offspring versus fostered birds.

To eliminate the confound of potential variability in individual interactions with the tutor, we next used a computer tutoring paradigm (*Mets and Brainard, 2018a*; *Tchernichovski et al., 1999*) to hold both the acoustic structure of the tutor song and the number of song exposures constant across all individuals (*Figure 3A*). We limited exposure to auditory stimuli other than the computer tutor song by transferring eggs within 36 hr of laying to nests where hatchlings were reared to independence by non-singing female foster parents. At ~45 days post-hatch, juveniles were transferred

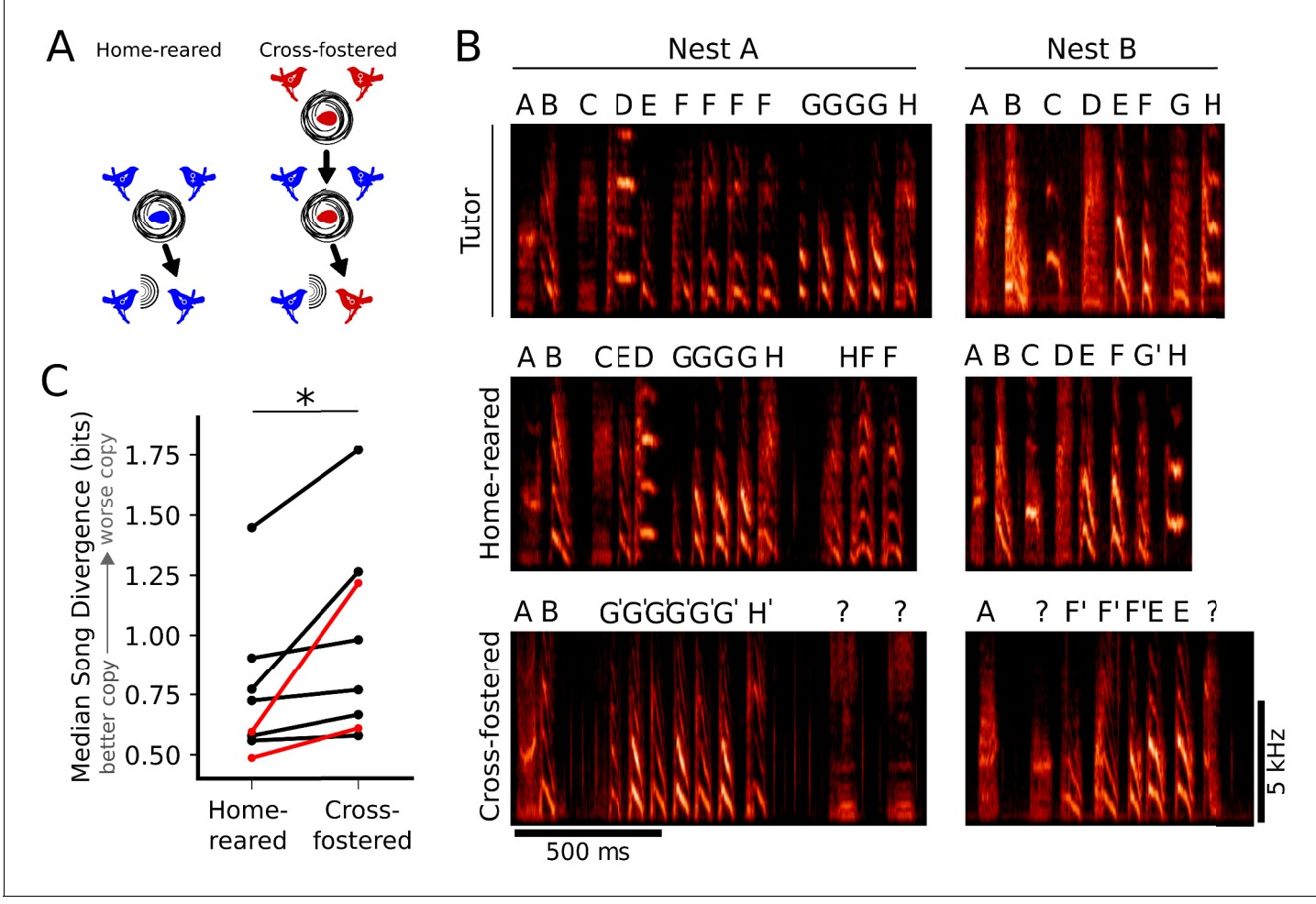

**Figure 2.** Learning outcomes are better when juveniles are tutored by their genetic fathers. (**A**) Schematic representation of the experimental design. Parental pairs (middle row, blue birds) reared both their own genetic offspring (home-reared, left), and foster birds from different genetic backgrounds (cross-fostered, right). (**B**) Examples of learning outcomes for two nests. (**B**, top) Example songs from the resident male tutors in each nest. (**B**, middle, bottom) Example songs from birds that were either home-reared (**B**, middle) or cross-fostered (**B**, bottom) in nests where males sang the tutor songs indicated in B, top. The examples of learned songs are from birds that had the median Song Divergence (SD) scores for their cohort (home-reared or cross-fostered). Syllable labels are provided to facilitate comparisons, but do not reflect automated SD scores used to quantify song similarity. (**C**) Paired plot of median SD scores for home-reared and cross-fostered cohorts. Across eight nests, birds that were home-reared learned significantly better than birds that were cross-fostered (n = 8 parental pairs, 52 cross-fostered birds, 45 home-reared birds; Wilcoxon signed-rank test, p<0.005). Median SD scores for nests corresponding to spectrograms in panel B are shown in red.

DOI: https://doi.org/10.7554/eLife.47216.004

The following figure supplement is available for figure 2:

**Figure supplement 1.** Pedigree of all tutor pairs used in cross-fostering experiments.

DOI: https://doi.org/10.7554/eLife.47216.010

to sound chambers where they were computer-tutored with identical songs, yielding a cohort of 20 birds from 13 different breeding nests, all with the same tutor song instruction.

Despite exposure to controlled computer tutoring experience, the distribution of learning outcomes was similar to that generated by live tutoring; some birds learned songs for which the spectral content of syllables closely resembled the tutor stimulus (*Figure 3B*, top row of 'tutee songs', and 3C) while other birds learned songs that had little resemblance to the tutor stimulus (*Figure 3B*, bottom row of 'tutee songs', and 3C). Thus, differences in parental behavior and in individual experience with the live tutor could not account for observed variation in the quality of learning.

In contrast, we found that a significant amount of variation in the quality of learning could be explained by how well the tutor stimulus matched individual genetic biases of juveniles. Previous

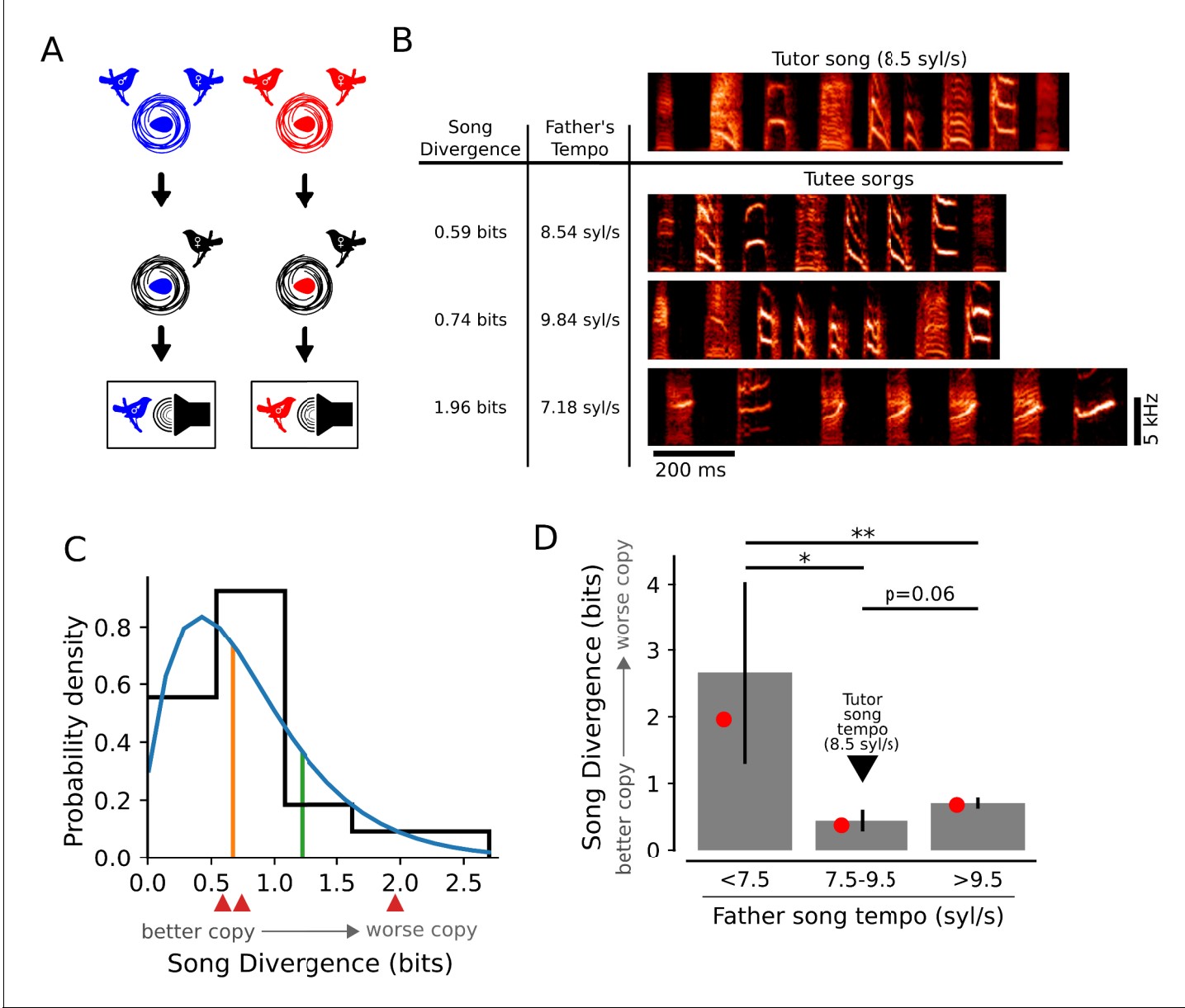

**Figure 3.** Learning outcomes are influenced by the relationship between tutor song tempo and genetic bias. (**A**) Schematic of computer tutoring. Individuals from different genetic backgrounds (red and blue nests) were raised by non-singing females to prevent exposure to their fathers' songs and were then tutored using computer-controlled playback of identical tutor stimuli. (**B**, top) Spectrogram of a full motif from the synthetic computer tutor song with a tempo of 8.5 syl/s. (**B**, bottom) Example songs for three tutees from different nests displaying variation in learning outcomes for song spectral content. The SD scores for the song repertoires of these individual birds are shown next to the example spectrograms. These tutees (from top to bottom) came from nests where the fathers' song tempos were similar to (8.5 syl/s), faster than (9.8 syl/s), and slower than (7.2 syl/s) the tutor song. (**C**) The distribution of learning outcomes for 20 birds drawn from 13 different genetic backgrounds. A gamma distribution fit to the data (blue line) and the 50th percentile (orange) and 80th percentile (green) indicate that learning outcomes were skewed towards better learning. Red arrows correspond to the learned songs presented in panel B. (**D**) Mean (gray bars) and median (red dots) SD scores for birds that were grouped by the tempos of their genetic fathers' songs. Learning was best for birds that had a genetic bias to sing near the tempo of the tutor song (fathers' song tempos from 7.5 to 9.5 syl/s), while learning was worse for birds that were biased to sing slower than the tutor song (<7.5 syl/s; * denotes p<0.02, one-tailed t-test) or faster than the tutor song (>9.5 syl/s; p=0.06, one-tailed t-test). Additionally, birds that were biased to sing faster than the tutor song learned better than birds that were biased to sing slower than the tutor song (** denotes p<0.02, two-tailed t-test). All p-values were corrected for multiple testing using the Holm-Bonferroni procedure.
DOI: https://doi.org/10.7554/eLife.47216.005

work (*Mets and Brainard, 2018a*) demonstrated that juveniles learn songs with tempos that are strongly biased towards the tempos of their fathers' songs, even when they have never heard their fathers sing. We therefore estimated the genetic bias for tempo of individuals from different nests as the tempo (in syl/s) measured for their fathers' songs. Across the juvenile birds that were computer tutored, the individual biases for tempo ranged from 5.5 syl/s to 12.5 syl/s. Hence, the computer tutor stimulus, which was presented at 8.5 syl/s, was better matched to the genetic biases of some individuals than for others. We measured the quality of learning for computer tutored birds in three groups: birds that had biases for tempo that were slower than (<7.5 syl/s), similar to (7.5–9.5 syl/s), or faster than (>9.5 syl/s), the tempo of the tutor song (8.5 syl/s). Across these groups, the birds that had biases for tempo that were most similar to the tutor song learned best (*Figure 3D*). Thus, as with live tutoring, the quality of learning for spectral content of a controlled, synthetic song could be explained in part by the alignment between instructive experience and individual genetic bias.

To more explicitly test whether matching the tempo of the tutor song to the genetic bias of individual birds could enhance learning, we carried out an additional computer tutoring experiment in which we assessed how varying the tempo of the tutor song influenced the quality of learning for birds from three different genetic backgrounds that were biased to sing at differing song tempos - one near the lower decile of song tempos produced in our colony (7.18 syl/s father song tempo), one near the median (7.89 syl/s), and one near the upper decile (10.31 syl/s). We tutored groups of juveniles from each of these 'slow', 'medium' and 'fast' genetic backgrounds with songs that had identical spectral content but that were presented at 'slow', 'medium' and 'fast' tempos (6.5 syl/s, 8.5 syl/s, and 10.5 syl/s; see Materials and methods) (n = 28 animals; Schematized, *Figure 4A*). Synthetic tutor songs that differed in tempo were constructed by varying the gaps between syllables, without altering either the durations or spectral content of the syllables themselves (*Figure 4—figure supplement 1*). Hence, every bird was tutored with stimuli that had the same spectral content and number of syllables. Correspondingly, in order to compare learning across groups tutored with different tempo songs, we used the Song Divergence, which quantifies how well syllable spectral content is learned independently of song temporal structure. The differences in quality of learning that we observed therefore reflect influences of the rate at which syllables are presented on an orthogonal aspect of song structure - how well the spectral content of syllables is learned.

Joint consideration of tutor song tempo and individual genetic bias revealed a strong interaction between the two; for each of the three genetic backgrounds, the best learning was achieved if tutor song tempo was 'matched' to the genetic bias (*Figure 4A,B*; example spectrograms illustrated in *Figure 4—figure supplement 1* and individual groups quantified in *Figure 4—figure supplement 2A*). Within the 'unmatched' groups, birds learned better if they were tutored with a song slower than their genetic bias (*Figure 4B*, p<0.02, two-tailed t-test, Holm-Bonferroni correction for multiple comparisons). These three categories (tutor song faster than father's song, tutor song matched to father's song, and tutor song slower than father's song) accounted for 39.8% of the variation in song learning outcomes (ANOVA, p<0.008, Holm-Bonferroni correction for multiple comparisons). In contrast, genetic background considered alone, and tutor song tempo considered alone, explained only 4.1% and 0.28%, respectively, of the variation in learning (*Figure 4—figure supplement 2B,C*). These findings demonstrate that much of the quality of song learning can be explained by the interaction between tutor song experience and genetic bias.

We considered whether the interaction between tutor experience and genetic bias could reflect a trade-off between learning song spectral structure and optimizing tempo. In particular, for some species, it has been suggested that individuals strike a balance between producing syllables with broadband spectral content and producing those syllables at a faster tempo (*Lahti et al., 2011*; *Podos, 1997*; *Podos et al., 2004*). This observation raises the possibility that worse SD scores in our study reflect in part a potentially advantageous sacrifice in the quality of spectral copying in order to optimize song tempo. To examine this possibility, we first tested whether SD scores were worse for individuals that sang faster songs than for individuals that sang slower songs, as might be expected if there was a trade-off between the quality of spectral copying and maximizing song tempo - a song feature that has been identified in some studies as more attractive to females and therefore favored (*Ballentine, 2004*; *Nowicki and Searcy, 2005*). We did not find this trend in our data (*Figure 4—figure supplement 3A*). We then tested whether SD scores were worse for individuals that sang songs more closely matched in tempo to the tutor song, as might be expected if there was a

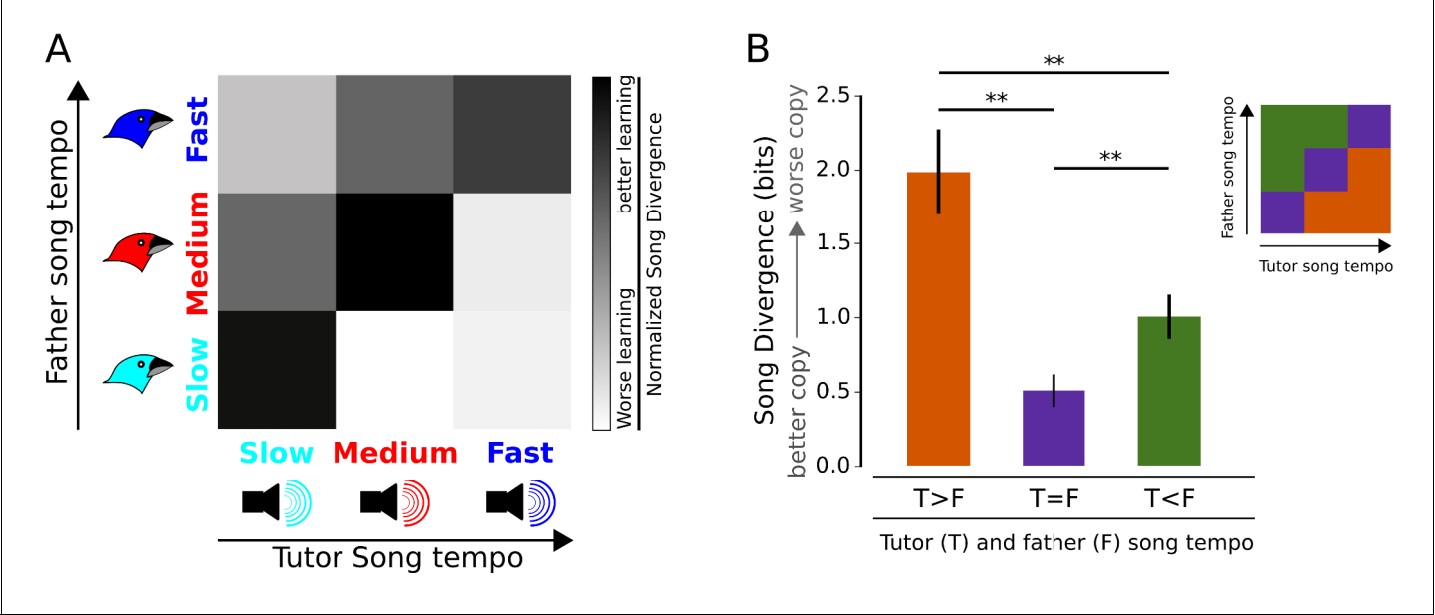

**Figure 4.** Matching tutor song tempo to the genetic bias of an individual results in better learning. (**A**) Cohorts of individuals from 'slow', 'medium' and 'fast' genetic backgrounds (left) were tutored with 'slow', 'medium' and 'fast' synthetic tutor songs (bottom). The normalized, average Song Divergence for each of the nine cohorts is indicated in grayscale, with darker shades indicating better learning. (**B**) Across the three genetic backgrounds, learning outcomes were best when tutor-song tempo and genetic bias were matched (purple vs. orange and green). If unmatched, tutor song was learned better if tutor song was slower than genetic bias for song tempo (orange vs. green). For all panels, error bars are standard error. ** denotes $p<0.02$, two-tailed t-test. All p-values were corrected for multiple testing using Holm-Bonferroni procedure.

DOI: https://doi.org/10.7554/eLife.47216.006

The following figure supplements are available for figure 4:

**Figure supplement 1.** Birds from a genetic background biased to sing at a medium song tempo learn best when provided with a tutor song at a medium song tempo.

DOI: https://doi.org/10.7554/eLife.47216.007

**Figure supplement 2.** Quantification of song learning for specific tutor-song tempo father-song tempo pairings.

DOI: https://doi.org/10.7554/eLife.47216.008

**Figure supplement 3.** Birds did not deviate from good syllable copying in order to optimize song tempo.

DOI: https://doi.org/10.7554/eLife.47216.009

trade-off between the quality of song spectral copying and the ability to match tutor song tempo. In contrast to this possibility, we found that there was significantly better copying of syllable spectral content in birds that also more closely matched song tempo to the tutor song (*Figure 4—figure supplement 3B*). This is consistent with the idea that the ability of an individual to match the tutor song tempo facilitates the learning of syllable spectral content, and argues against the possibility that birds with poor learning of spectral content are optimizing some other song feature.

## Discussion

Together, the cross-fostering and computer tutoring experiments show that tailoring an instructive stimulus to match the genetic bias of an individual can enhance learning. It is noteworthy that our estimates of individual bias derived exclusively from the tempo of the father's song; hence, a more accurate estimate of bias, incorporating other genetic factors such as maternal contributions, would potentially enable even more effective tailoring of instruction. Our results additionally indicate that a failure to take into account how experience and genetics interact can lead to erroneous conclusions about the immutability of genetic constraints on individual differences in learning. For example, absent consideration of this interaction, birds that are genetically biased to sing slow songs appear to be inherently worse learners than birds that are biased to sing fast (e.g. averaged across all stimuli, slow birds learn worse than fast birds; see *Figure 4—figure supplement 2*). However, when

instruction is individually tailored, it is apparent that 'slow' birds can learn as well as, or better than, 'fast' birds (*Figure 4A* and *Figure 4—figure supplement 2*; slow birds tutored with slow songs learned better than fast birds tutored with fast songs). Hence, customization of instruction can not only improve learning outcomes, but in so doing may also attenuate differences across individuals that otherwise might have been construed as genetically determined.

These results also inform an understanding of how genetic factors shape the cultural transmission of song within a population of birds. Previous work has indicated that songbirds are subject to genetic constraints at the species level that bias birds to learn better from songs that are more species-typical (*Gardner et al., 2005*; *Lahti et al., 2011*; *Marler and Peters, 1988*; *Podos, 1997*; *Podos, 1996*; *Podos et al., 2004*), and it has been proposed that such constraints might contribute to the long-term stability of a given species' song. According to this idea, learning drives variation in songs across individuals, but genetic factors 'pull' all birds back toward a single species-specific song 'template', thus reducing drift in the population level song structure over generations (*Fehér et al., 2009*; *Lachlan et al., 2018*; *Lachlan and Slater, 1999*; *Lynch, 1996*).

The observation that individual Bengalese finches are biased to learn better from some songs than from others is broadly consistent with the idea that innate factors might constrain 'drift' away from species-typical song models. However, our finding that within this genetically heterogeneous population, each bird or nest may have a different genetic bias, indicates that there is no single song model toward which all individuals are drawn (indeed, the median song tempo in our colony was 8.5 syl/s, yet this 'species typical' song is a poor model for birds from faster and slower nests). Rather, our findings suggest an alternative possibility - that whatever genetic biases toward different song structure are present between nests or families could be preserved or amplified across generations, potentially contributing to gradual divergence of song structure within distinct subpopulations of birds.

These results also demonstrate that learning of complex skills like song can be shaped by both direct and indirect influences of parental genes. For home-reared birds, genetic factors bias the acoustic structure of the father's song, including tempo, but also contribute to a similar bias in his offspring (*Mets and Brainard, 2018a*); this establishes an alignment of experience and genetics that we have shown can enhance learning. These findings support the idea that, for human families, a similar synergistic interaction between the home environments that are shaped by parental behavior, and the heritable predispositions of their children, contributes to the observed clustering within families of achievement in verbal, quantitative, musical and athletic domains (*Meredith, 1973*; *Plomin et al., 1977*; *Tan et al., 2014*; *Vinkhuyzen et al., 2009*). Such interactions may be especially potent for behaviors like speech and other motor skills, where 'tutoring' in the early home environment plays a critical role in shaping the perceptual and motor systems underlying performance (*Kuhl, 2010*).

More broadly, our findings highlight the critical role that gene-experience interactions can play in determining complex phenotypes. When considered alone, neither genes nor experience had a clear impact on song learning, but their interaction explained nearly 40% percent of variation in learning outcomes. Detection of this strong interaction required detailed knowledge of individual genetic predispositions and experience, information that is often hard to obtain in the context of human studies (*Halldorsdottir and Binder, 2017*). Nevertheless, a better understanding of such interactions between genetics and experience will likely be required to fully elucidate the mechanisms driving individual-to-individual variation in complex learned behaviors.

## Materials and methods

### Subjects

Subjects were male Bengalese finches (*Lonchura striata domestica*). 239 birds were reared and tutored by live birds and 47 additional birds were reared by foster females and tutored by computer. These animals were bred in our colony. Other than efforts to maintain some separation between lineages, mating pairs comprised randomly selected male and female birds. All protocols were reviewed and approved by the Institutional Animal Care and Use Committee at the University of California, San Francisco. Data on tempo from a subset of the cross-foster and computer tutored birds was presented in a previous study (*Mets and Brainard, 2018a*).

## Audio recording and initial processing

For audio recording, birds were single-housed in sound isolation chambers (Acoustic Systems). Songs were digitally recorded at a sampling frequency of 32 kHz, and a bit depth of 16. Recording microphones were placed in a fixed position at the top of the cage housing the bird. Prior to further analysis, all songs were high-pass filtered at ~500 Hz using a digitally implemented elliptical infinite impulse response filter with a passband edge frequency of 0.04 radians. All recordings used for analysis were acquired during early adulthood (90–120 days post hatch).

## Cross-fostering

To create populations of birds that were tutored by a live bird but never heard the song of their genetic fathers, eggs were taken from parents within 36 hr of laying and transferred to foster nests where hatchlings were reared to adulthood. For each individual, the specific foster nest was randomly selected from a set of 8 foster nests within our breeding colony. All birds that were studied in these experiments were offspring of breeding pairs from our large and genetically heterogeneous colony. The breeding pairs themselves were established from birds that were acquired over a multi-year period from outside vendors, or were bred in house (see for example *Figure 2—figure supplement 1*). Given the structure of our colony, the cross-fostered birds were less related to their tutors than were the home-reared birds, but in many cases, the cross-fostered birds were more or less distant 'cousins' of their tutors. Our expectation is that effects we reported for these experiments (consistently better learning for genetic-offspring than for less-closely related cross-fostered birds) would have been even more pronounced had the cross-fostered birds been drawn from entirely different colonies or other outside sources.

## Quantification of song learning

The quality of song learning was measured using Song Kullback-Leibler Divergence or Song Divergence (*Mets and Brainard, 2018b*). This is a largely automated measure that estimates the degree to which the spectral content of a tutor song is copied by a tutee, and has been shown to have a good correspondence with human assessments of song learning (*Mets and Brainard, 2018b*). Song Divergence is computed by creating a statistical model that captures the song spectral content of both tutor and tutee and estimating the divergence between these two models. This estimate of divergence is directional; the Song Divergence calculated in the tutor-tutee direction indicates the amount of song spectral content (in bits) present in the tutor song that is not present in the tutee song ("missing content") but does not capture spectral content from the tutee song that is not present in the tutor song ("improvisation"). The model for each bird in our study was generated from a corpus of 60 song bouts as follows. Songs were segmented into discrete units of sound (syllables) separated by silence using an automatically determined amplitude threshold. For each syllable, the spectral content was extracted by calculating a power spectral density (PSD). Here, we used a single PSD per syllable, removing information about syllable temporal structure. Compared to other semi-automated methods for evaluating song learning, this allows assessment of the copying of song spectral content independently of song temporal structure (*Burkett et al., 2015*; *Mandelblat-Cerf and Fee, 2014*; *Mets and Brainard, 2018b*; *Tchernichovski et al., 2000*). These PSDs were transformed into a syllable-syllable similarity space and a Gaussian mixture model (GMM) was then fit to the distributions of these syllables. The Kullback-Leibler Divergence between two GMMs corresponding to songs from a tutor and a tutee is the Song Divergence. In the case of a synthetic tutor stimulus where there is only a single variation of the song, Song Divergence was calculated by comparing the song corpus of a tutee to the song corpus of a bird that had learned the tutor song especially well. The song corpus of the same 'tutor' bird was used for all such calculations.

## Song tempo calculation

Song tempo was quantified as the average number of syllables produced per second of song, a measure we have previously used to identify song tempo as heritable (*Mets and Brainard, 2018a*). Discrete units of sound separated by silence (syllables) were identified based on amplitude. First, an 'amplitude envelope' was created by rectifying the song waveform and then smoothing the waveform through convolution with an 8 ms square wave. Threshold crossings of this amplitude trace were then used to identify periods of vocalization. Thresholds were set heuristically to result in

segmentation that corresponded with syllable onsets and offsets apparent in human examination of spectrograms. Once the threshold was established, 'objects' were identified as uninterrupted regions longer than 10 ms over which the amplitude envelope exceeded threshold. Any objects separated by a gap of 5 ms or less were merged producing a final set of objects that were defined as syllables. A series of syllables that had no gaps larger than 250 ms was considered a song bout. For each bird, tempo was then quantified as the number of syllables present in a song bout divided by the duration of the song bout, averaged across at least 60 bouts of singing.

## Computer tutoring

To create populations of birds that had controlled tutoring experience, eggs were taken from parents within 36 hr of laying, prior to neural development (*Murray et al., 2013*; *Yamasaki and Tonosaki, 1988*), and were then raised by pairs of non-singing foster mothers housed in sound isolation chambers. While the female foster mothers do not sing, they produce unlearned calls and influence other aspects of a hatchling's experience, including feeding and social interactions, that could plausibly influence hatchling development and the quality of subsequent computer driven song learning. In order to ensure that any such influences of female fostering on song learning did not contribute systematically to our results, we used 14 different female foster nests (each with two foster mothers) and randomly assigned eggs from experimental nests to these foster nests. In order to prevent any 'batch effects', only a single hatchling was raised at a time in any given foster nest. Foster mothers raised the juveniles until they were able to feed themselves. At independence (usually 35–40 days post hatch) birds were moved to an acoustic isolation chamber with an audio recording system and a computer tutoring apparatus, based on an approach that previously has been demonstrated to drive song learning (*Mets and Brainard, 2018a*; *Tchernichovski et al., 1999*). At 45 days post-hatch, the tutoring apparatus was activated, allowing birds to access a tutor song. The apparatus consisted of a perch activated switch that caused playback of a tutor stimulus (see below). Each perch hop elicited a single playback of the tutor stimulus. Birds were allowed to playback 10 songs, three times a day (morning, noon, and evening). Playback of tutor song was limited to 30 songs per day based on previous work indicating that this was near an optimal value to maximize the quality of song learning in this paradigm (*Tchernichovski et al., 1999*). It sometimes took birds a few days to begin consistently actuating song playbacks. Nevertheless, all birds elicited at least 90% of the available song playbacks during the tutoring period (median = 96%), and there was no relationship between the number of playbacks that an individual heard and the quality of song learning. The computer tutoring apparatus was implemented with custom LabView software (National Instruments, "EvTutor") that is provided in supporting materials for this paper. Birds remained in the tutoring apparatus until 120 days post-hatch. For experiments involving different tutor song tempos, the tempo for an individual was randomly selected from three possible tempos (see below).

## Computer tutor stimulus

The synthetic stimulus used during playback tutoring was the same as that used in our previous work (*Mets and Brainard, 2018a*; *Mets and Brainard, 2018b*). To create a naturalistic but controlled learning stimulus, a synthetic song used for computer tutoring was derived from songs sampled from our Bengalese finch colony. The synthetic song was composed of 9 categorically distinct syllables that were chosen to reflect a range of different syllable types found in Bengalese finch song (i.e. short 'introductory' syllables, noisy syllables, syllables with harmonic structure and constant or modulated frequency, etc.). The tutor stimulus consisted of a series of introductory syllables followed by three repetitions of a stereotyped sequence of syllables, or 'motif' (shown in *Figure 3B*). The gaps between syllables were chosen to reflect naturalistic means and standard deviations based on the distribution of gap durations found in normal Bengalese finch song. Correspondingly, the tutor song stimulus had a relatively natural prosody compared to a stimulus in which the time between syllable onsets is fixed. The 8.5 syl/s tutor song stimulus arose naturally out of this process, as 8.5 syl/s is close to the median song tempo present in our colony. The 6.5 and 10.5 syl/s tutor stimuli were created by proportionally increasing or decreasing only the inter-syllable gap durations, resulting in songs with identical spectral content presented at different tempos.

## Statistics

All statistical testing in this study was carried out in consultation with the biostatistics consultancy of the UCSF Clinical and Translational Science Institute. No birds were removed from the study. As no subjective measurements were made, no blinding was performed. When used, statistical tests were appropriate to the data presented and the data, to the limit of detection, were consistent with the assumptions of the tests. For all ANOVA analyses, estimates of variance explained are Omega squared. For each experimental group, p-values for statistical tests were corrected for multiple testing within that group using the Holm-Bonferroni procedure, an extension of the Bonferroni correction which retains the same family-wise error rate while reducing the false negative rate relative to traditional Bonferroni (*Holm, 1979*). Unless otherwise indicated, all tests were two-tailed. For all cases, where a one-tailed test was conducted because of a directional hypothesis, the threshold p-value for the test was considered to be significant at the 0.025 level to reduce the false positive rate to be equivalent to a two-tailed test. For all tests, the statistical significance was not impacted by the use of a one-tailed versus a two-tailed test. For the cross-fostering experiments presented in *Figure 2*, the median within-nest SD scores for home-reared birds were compared to the median within nest SD scores for cross-fostered juveniles. Median values were used as a summary statistic for this comparison because the SD scores were approximately gamma distributed (*Figure 1C*). The effect of home-rearing on song learning was tested using a Wilcoxon signed-rank test, a paired test which tests the sign of the change across experimental conditions and not the magnitude of the change. This enables a test for the impact of home-rearing relative to cross-fostering while controlling for nest-specific or tutor-song-specific variables.

## Acknowledgements

The authors thank A Karpova, and W H Mehaffey for discussions and comments on the manuscript. This work was supported by the Howard Hughes Medical Institute, a PBBR award from the Sandler Family Foundation (MSB), and The Jane Coffin Childs Fund for Medical Research (DGM).

## Additional information

### Funding

| Funder | Grant reference number | Author |
|---|---|---|
| Jane Coffin Childs Memorial Fund for Medical Research | Fellowship to DGM | David G Mets |
| Howard Hughes Medical Institute | | Michael S Brainard |
| Sandler Foundation | PBBR award | Michael S Brainard |

The funders had no role in study design, data collection and interpretation, or the decision to submit the work for publication.

### Author contributions

David G Mets, Conceptualization, Data curation, Software, Formal analysis, Funding acquisition, Investigation, Visualization, Methodology, Writing—original draft, Project administration, Writing—review and editing; Michael S Brainard, Conceptualization, Supervision, Funding acquisition, Writing—original draft, Writing—review and editing

### Author ORCIDs

David G Mets (ID) https://orcid.org/0000-0002-0803-0912
Michael S Brainard (ID) https://orcid.org/0000-0002-9425-9907

## Ethics

Animal experimentation: All of the animals were handled according to approved institutional animal care and use committee (IACUC) protocols (#AN170723-02) of the University of California, San Francisco.

## Decision letter and Author response

Decision letter https://doi.org/10.7554/eLife.47216.013
Author response https://doi.org/10.7554/eLife.47216.014

## Additional files

### Supplementary files

• Source code 1. Tutoring software.
DOI: https://doi.org/10.7554/eLife.47216.011

### Data availability

All data generated or analysed during this study are included in the manuscript and supporting files.

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
