## [Decision Letter]

Thank you for submitting your article "Learning is enhanced by tailoring instruction to individual genetic differences" for consideration by *eLife*. Your article has been reviewed by three peer reviewers, and the evaluation has been overseen by Ronald Calabrese as the Senior and Reviewing Editor. The following individuals involved in review of your submission have agreed to reveal their identity: Marc F Schmidt (Reviewer #2); Frank Johnson (Reviewer #3).

The reviewers have discussed the reviews with one another and the Reviewing Editor has drafted this decision to help you prepare a revised submission.

Summary:

In a previous study (PNAS 2018) the authors showed that genetic predispositions for vocal learning can be demonstrated in songbirds when birds are tutored using computerized instruction but that this difference can be eliminated when birds are tutored with live tutors. In this study the authors show that tailoring tutoring to the type of genetic predisposition can overcome potential deficits in learning. The authors further show that matching the tempo of the tutor song with the tempo of the tutee's father song (i.e. the genetically preferred tempo) greatly improves the tutee's overall ability of learning the acoustic features of the song. Tutoring juveniles with songs that are either of a faster or slower tempo than the father's song causes a decrease in overall copying performance when measured as a degree of acoustic similarity of individual syllables.

Essential revisions:

The results are exciting and of potential general interest, however, there were some concerns that must be addressed before publication. The written reviews are attached and are not completely consistent but do contain valuable insight into details of any revision plan. In consultation the reviewers reached the following consensus.

1) Given that authors recently shown that song tempo is heritable, and that other previous work showed that rhythm mismatch can decrease imitation (a 'calibration effect'), the current manuscript must more clearly articulate the novelty of the results. It seems reasonable to conclude that tutoring birds with songs that are faster in tempo than their predisposed (genetically determined) tempo, should have a negative effect on song learning. A similar effect was documented by Podos and Nowicki, 1999, who showed a calibration effect and 'broken syntax' (deviation from tutor song) when tutor song rhythm was too fast. In other words, similarity to tutor song decreases not necessarily because the bird 'failed' to imitate, but because the bird might have actively deviated from the tutor in order to optimize performance. Authors could test for such a scenario in their current data, and minimally they must discuss such an alternative scenario and clarify how their results are novel.

2) There were concerns about the statistical methods. First, in the operant training group authors performed more than one statistical test, but did not account for multiplicity, which pose a risk of false discovery. Further, in one of those tests (Figure 4B), authors report one-tailed t-test, which seems unjustified (one-tailed should only be used in cases where it is impossible to obtain the other tail, which is not the case here if we understand the measures correctly). Given that the one-tailed test gave p<0.02, and that multiple comparisons were not accounted for, the correct p-value should be 0.08.

3) The claims about genetic differences are based a common garden approach which is standard for determining heritability and therefore seems appropriate, and the overall design, was able to remove all possible environmental influences on learning. Nevertheless, there were concerns about the claims for genetic influence that are detailed in the comments of reviewer #3. The authors need not implement further genetic studies but two changes must be implemented to give more confidence in the conclusions. First a pedigree analysis of the birds used in the study should be provided in supplementary figures and claims about genetic difference and the degree of genetic variation in the population should be tempered in light of these pedigrees. Second, while in consultation it was agreed among the reviewers that the authors have done a good job controlling for maternal influence and thus showing that it is not necessary for the observe effects, the possibility of a maternal influence should be discussed.

*Reviewer #1:*

This study tested for heritable predispositions in song learning outcomes in Bengalese finches. Results show that chicks raised by foster parents imitated their father song less accurately compared to biological offspring. Interestingly, the decrease in song learning can be explained, to a large extent, by differences in tutor's song tempo, a result that authors confirmed in controlled operant song tutoring experiments.

Overall, this is an interesting and well executed study. However, given that authors have recently shown that song tempo is heritable, and that other previous studies showed that rhythm mismatch can decrease imitation (a 'calibration effect'), I found it difficult to evaluate the novelty of the results. Overall, it seems very reasonable to conclude that tutoring birds with songs that are faster in tempo than their predisposed (genetically determined) tempo, should have a negative effect on song learning. A similar effect was documented by Podos and Nowicki, who showed a calibration effect and 'broken syntax' (deviation from tutor song) when tutor song rhythm was too fast. In other words, similarity to tutor song decreases not necessarily because the bird 'failed' to imitate, but because the bird might have actively deviated from the tutor in order to optimize performance. Authors should test for such a scenario in their current data.

Finally, I had a few issues with the statistics: first, in the operant training group authors performed more than one statistical test, but did not account for multiplicity, which pose a risk of false discovery. Further, in one of those tests (Figure 4B), authors report one-tailed t-test, which seems unjustified (one-tailed should only be used in cases where it is impossible to obtain the other tail, which is not the case here if I understand the measures correctly). Given that the one-tailed test gave p<0.02, and that multiple comparisons were not accounted for, the correct p-value should be 0.08.

*Reviewer #2:*

In a previous study (PNAS 2018) the authors showed that genetic predispositions for vocal learning can be demonstrated in songbirds when birds are tutored using computerized instruction but that this difference can be eliminated when birds are tutored with live tutors.

In this study the authors show that tailoring tutoring to the type of genetic predisposition can overcome potential deficits in learning.

In this study the authors show that matching the tempo of the tutor song with the tempo of the tutee's father song (i.e. the genetically preferred tempo) greatly improves the tutee's overall ability of learning the acoustic features of the song. Tutoring juveniles with songs that are either of a faster or slower tempo than the father's song causes a decrease in overall copying performance when measured as a degree of acoustic similarity of individual syllables.

The paper is extremely well written and clear. The flow is logical and the findings have potentially profound implications on motor skill learning more generally by suggesting that tailored instruction can overcome potential genetic biases.

The authors arguments for a genetic predisposition in learning ability is based on a common garden approach which is standard for determining heritability and therefore seems appropriate. The overall experimental approach is elegant in that juvenile males are isolated from the father at a very early age (36 hours post hatch) significantly before the auditory system fully develops and therefore before they would get the opportunity to hear their fathers' song. In most of the experiments, juveniles are then raised with a "neutral" female before being tutored (at 45 days of age) using a computer-based system that is able to deliver songs at different tempos. The overall design, in my opinion, therefore is able to remove all possible environmental influences on learning and can as such make a strong claim regarding heritability influences.

I do not have any major criticisms of the paper

*Reviewer #3:*

This fascinating manuscript addresses the interaction of genetic predisposition and experience in determining adult behavioral phenotypes. However, in my view, needed data are lacking to justify the strong claims made about genetics in driving behavioral outcomes.

For example, in the absence of a pedigree chart it is difficult to assess the apparent father-to-son heritability of a genetic predisposition to learn songs of a specific temporal structure. I am puzzled by the absence of any consideration of a maternal contribution to a genetic predisposition to song imitation – perhaps I have misunderstood, but this seems like an issue that the authors should address or acknowledge. A pedigree chart could certainly help sort this out.

The role of genetic factors would also be strengthened by a multi-generational experiment showing the persistence of the genetic predisposition. While the manuscript's single-generation experiments are certainly compelling and clearly sufficient to demonstrate learning, a defining characteristic of a genetic influence is persistence across generations.

Because I am enthusiastic about the research direction described in the manuscript, I would be happy to be corrected if the authors believe that I am in error. Comments regarding specific sections of the manuscript are listed below.

"In agreement with extensive prior research, we found that there was a broad range in the quality of song learning across individuals in our genetically heterogeneous Bengalese finch colony."

Could the authors provide references for "extensive prior research" on individual variation in song learning? A number of references are cited in the preceding paragraph, but these papers refer primarily to species or strain differences.Could the authors define what they mean by "genetically heterogenous?" Are there breeding records that establish the degree of genetic heterogeneity within the colony? This seems important, given the role that genetic predisposition is argued to play in the analysis of the findings.

"Prior work indicates that the broad variation in the quality of song learning (Figure 1C) could be influenced by both experiential and genetic factors."

• Seems like there are additional references that could be included here on the role of experience.

"We compared learning for birds that were home-reared and tutored by their genetic fathers, to learning for birds that were cross-fostered and tutored by genetically unrelated adults."

• What does "genetically unrelated" mean here? Were steps taken to insure a specific genetic distance between the adult pairs used in the home-reared and cross-fostered conditions?

Figure 2C: The effect shown in Figure 2C appears to be driven primarily by 3 of the 8 nests.

• Could the authors justify or explain use of a signed-rank test to determine significance, as well as use of median SD scores?

• Throughout the manuscript, reporting the results of statistical tests is reduced to the name of the test and a p-value. Not sure if this is journal preference but seems non-standard.

"Despite exposure to identical computer tutoring experience"

• In the Materials and methods subsection “Computer tutoring”, the authors describe their perch-activated tutoring setup. Because tutor-song playback was contingent on the tutee hopping on the perch, one wonders if all birds were equally active in terms of their perch hopping. Although the number of tutor song playbacks per day was limited to 30, were there days when some birds received fewer because of less perch hopping? It would be helpful if the authors provided some evidence to support the statement that all birds received identical computer tutoring experience.

"Thus, differences in parental behavior and in individual experience with the live tutor could not account for observed variation in the quality of learning."

• Seems like a nest-level analysis would be useful here. That is, the birds were raised to 35-40 days post-hatch by female foster parents. The text of the manuscript seems to assume that rearing by all female foster parents was of equivalent quality. A nest-level analysis of learning outcomes would address this assumption.

Figure 3B: Y-axis.

• Perhaps indicate "Mean Song Divergence (bits)" on the y-axis, since median is used in Figure 2C?

---

## [Author Response]

Essential revisions:The results are exciting and of potential general interest, however, there were some concerns that must be addressed before publication. The written reviews are attached and are not completely consistent but do contain valuable insight into details of any revision plan. In consultation the reviewers reached the following consensus.1) Given that authors have recently shown that song tempo is heritable, and that other previous work showed that rhythm mismatch can decrease imitation (a 'calibration effect'), the current manuscript must more clearly articulate the novelty of the results. It seems reasonable to conclude that tutoring birds with songs that are faster in tempo than their predisposed (genetically determined) tempo, should have a negative effect on song learning. A similar effect was documented by Podos and Nowicki, 1999, who showed a calibration effect and 'broken syntax' (deviation from tutor song) when tutor song rhythm was too fast. In other words, similarity to tutor song decreases not necessarily because the bird 'failed' to imitate, but because the bird might have actively deviated from the tutor in order to optimize performance. Authors could test for such a scenario in their current data, and minimally they must discuss such an alternative scenario and clarify how their results are novel.

As requested, we have carried out new analyses to test for the specific possibility raised here that poor copying of spectral structure might not reflect a failure to learn (because of a mismatch between genetic bias and tutor song tempo), but rather might reflect an active deviation from good spectral copying in order to optimize temporal aspects of song structure. The analyses that we carried out, detailed below and presented in revised text (Results, last paragraph) and Figure 4—figure supplement 3, argue against the possibility that the effects of genetic bias on learning that we observe could be explained by the kind of ‘trade-off’ or ‘calibration’ effects that are raised in Podos’s work:

As noted by the reviewers, work from Podos and colleagues raises the possibility that within a species there may be a trade-off between optimizing different features of song. For example, female birds might prefer both faster songs and songs with broader spectral bandwidth (and higher levels of spectral complexity), so that male birds have incentive to optimize both of these features. However, in this model, limitations on song production mechanisms create constraints that result in a trade-off between producing optimal values of these features. In support of this possibility, Podos, 1997, reported that within a given species, birds that produce the fastest songs generally have narrower spectral bandwidths, while birds that produce the slowest songs have the broadest spectral bandwidths; they further suggested that birds could 'calibrate' their songs at different positions along this tempo-spectral bandwidth axis, according to their individual capabilities, in order to optimize the learned song (Podos et al., 2004).

These observations from other species raise the possibility that birds in our study might not be failing to copy song spectral content (when tempo is mismatched to individual genetic bias), but, instead, might be actively “choosing” to produce worse spectral copying in order to optimize aspects of song tempo.

We therefore tested two aspects of song tempo that our birds might have been optimizing at the expense of copying song spectral content. First, we tested the possibility raised by Podos and colleagues (Lhati et al., 2011; Podos, 1996; Podos, 1997; Podos et al., 2004) that there is a tradeoff between producing better spectral content of syllables and producing faster songs (as faster songs in some species are construed as more attractive to females). According to this possibility, we would expect that better spectral copying would be associated with slower song tempos, and worse spectral copying with faster song tempos. We found no suggestion of such a relationship in our data (Figure 4—figure supplement 3A).

Second, we tested the possibility that there is a trade-off between better spectral copying and producing songs that match more closely to the tutor song tempo. According to this possibility, we would expect that better spectral copying would be associated with birds that produced a worse match to the tutor song tempo, and worse spectral copying would be associated with birds that produced a better match to the tutor song tempo. In contrast to such a trade-off between matching tutor song spectral content and tutor song tempo, we found the opposite result: birds with learned song tempos that most closely matched the tutor song tempo also produced better copying of spectral content (Figure 4—figure supplement 3B). This result argues against a ‘calibration’ or ‘trade-off’ effect and in favor of the interpretation that the capacity of individual birds to learn spectral content is enabled by a closer match of individual genetic predisposition to the tutor song tempo. These analyses are presented in the main text (Results, last paragraph) and Figure 4—figure supplement 3B.

In addition to addressing the specific issue of ‘calibration’ raised here, we have further elaborated in several places the broader significance of our observations regarding the interaction between individual genetic variation and experience in shaping learning and the “cultural transmission” of song and other complex phenotypes. We particularly emphasize the relevance of our findings to understanding how customization of instruction based on heritable differences across individuals can enhance learning. We also further discuss how the propensity to learn better from some songs (such as that of the father) over others, differs from a common presumption that there is “neutrality” of tutor song stimuli across a broad range of “species typical” songs. The concept of neutrality of tutor songs within a broad species-typical range has been incorporated into modelling of interactions of genes and experience in shaping cultural transmission and speciation (Lachlan, 1999; Lachlan et al., 2018; Lynch, 1996). A general idea that arises in this work is that the presence of a single, species-typical ‘template’ could prevent the drift of song structure over generations that might otherwise arise due to the effects of learning on individuals’ songs. That is, while learning can drive individual variation in song structure, for example across different families, species level genetic constraints ‘pull’ songs back towards species-typical values. In contrast to this prior work, our results show that within a species there may be no single “species-typical” template, and we have added a more explicit discussion of the possibility that individual and familial variation in genetic predispositions within a species could potentially contribute to divergence of song structure across subpopulations within a species (Discussion, second paragraph).

2) There were concerns about the statistical methods. First, in the operant training group authors performed more than one statistical test, but did not account for multiplicity, which pose a risk of false discovery. Further, in one of those tests (Figure 4B), authors report one-tailed t-test, which seems unjustified (one-tailed should only be used in cases where it is impossible to obtain the other tail, which is not the case here if we understand the measures correctly). Given that the one-tailed test gave p<0.02, and that multiple comparisons were not accounted for, the correct p-value should be 0.08.

We have revised our statistical testing in consultation with the biostatistics service of the UCSF Clinical and Translational Science Institute. In an expanded “Statistics” subsection of Materials and methods, we provide a more detailed description of statistical testing, and include additional information regarding directionality of hypothesis testing and corrections for multiple comparisons in the main text and figure legends.

We previously used one-tailed tests in instances where there was a directional hypothesis based on prior results. For example, we had hypothesized prior to the conduct of the cross-fostering experiments that home-reared birds would learn better than cross-fostered birds (rather than that they simply would be different, without regard to directionality). Similarly, for the computer tutoring experiments, we hypothesized that birds would learn better from tutor songs that had tempos matched to their genetic bias than from tutor songs that were unmatched. Our statistical consultancy endorsed the view that in cases such as this, where there is a directional hypothesis, one-tailed tests are appropriate. However, there is not universal agreement about what is the best practice, and we have therefore adopted the reviewer’s suggestion that we use more conservative two-tailed tests for the data in Figure 4B. Regarding multiple comparisons, we now use Holm-Bonferroni corrected p-values for statistical tests in all cases. Use of two-tailed tests versus one-tailed tests, and corrections for multiple comparisons, did not alter the statistical significance of any results of the study,

With respect to the results of Figure 4B: The p values for the one-tailed tests in Figure 4B that we previously reported as p < 0.02, had specific values of p = 0.006 (for comparison of F = T vs. F > T), p = 0.000004 (for comparison of F = T vs. F < T), and p = 0.0032 (for comparison of F < T vs. F>T). All three tests remain significant at p < 0.02 after switching to two-tailed tests and correcting for multiple comparisons using a Holm-Bonferroni correction.

3) The claims about genetic differences are based a common garden approach which is standard for determining heritability and therefore seems appropriate, and the overall design, was able to remove all possible environmental influences on learning. Nevertheless, there were concerns about the claims for genetic influence that are detailed in the comments of reviewer #3. The authors need not implement further genetic studies but two changes must be implemented to give more confidence in the conclusions. First a pedigree analysis of the birds used in the study should be provided in supplementary figures and claims about genetic difference and the degree of genetic variation in the population should be tempered in light of these pedigrees. Second, while in consultation it was agreed among the reviewers that the authors have done a good job controlling for maternal influence and thus showing that it is not necessary for the observe effects, the possibility of a maternal influence should be discussed.

Regarding ‘relatedness’:

Per the request of the reviewers, we have now included a pedigree in Figure 2—figure supplement 1 that illustrates what we know about the relatedness of birds used in the cross-foster study. For the 8 foster nests, we show the origins of both the resident male (tutor) and female traced back to 2001. These data illustrate that the relevant birds were dispersed across different lineages in our colony (in many cases derived from different vendors), but that in a number of instances, birds are more or less distantly related ‘cousins’, and we now provide this characterization in the Materials and methods (subsection “Cross-fostering”). We have also eliminated our too loose usage of the term ‘unrelated’ and instead referred to the relevant populations as ‘home-reared’ and ‘cross-fostered’ (Results, second and third paragraphs).

We also now note the important point that any relatedness between cross-fostered birds and tutors in these experiments would be expected to attenuate the effects that we report (rather than artifactually magnify them). That is, we are testing the hypothesis that shared genes between tutor and offspring, and correspondingly shared predispositions for producing songs with particular structure, enhance learning of the tutor’s song by his offspring, relative to the learning of the same tutor’s song by less closely related cross-fostered birds. To the extent that cross-fostered birds are more closely related to their foster tutors, we would expect that differences in the quality of learning for cross-fostered birds and genetic offspring would be smaller. Conversely, to the extent that we had studied cross-fostered birds that were more distantly related to their tutors (for example derived from other BF colonies) we would expect that our reported effects would be even larger (subsection “Cross-fostering”).

Regarding maternal influences:

With respect to the significance of our reported results, we note that this is a case where any maternal influences on heritable predispositions that we did not take into account would have tended to contribute noise to our estimates of heritability and would therefore have tended to ‘work against us’. Conversely, if we were able to improve our estimates of heritable predispositions by taking into account maternal contributions, we would correspondingly expect that we could explain even more variation in the quality of learning across individuals.

We now have added a brief discussion of these points: “It is noteworthy that our estimates of individual bias derived exclusively from the tempo of the father’s song; hence, a more accurate estimate of bias, incorporating other genetic factors such as maternal contributions, would potentially enable even more effective tailoring of instruction”.

Reviewer #1:[…] Overall, this is an interesting and well executed study. However, given that authors recently shown that song tempo is heritable, and that other previous studied showed that rhythm mismatch can decrease imitation (a 'calibration effect'), I found it difficult to evaluate the novelty of the results. Overall, it seems very reasonable to conclude that tutoring birds with songs that are faster in tempo than their predisposed (genetically determined) tempo, should have a negative effect on song learning. A similar effect was documented by Podos and Nowicki, who showed a calibration effect and 'broken syntax' (deviation from tutor song) when tutor song rhythm was too fast. In other words, similarity to tutor song decreases not necessarily because the bird 'failed' to imitate, but because the bird might have actively deviated from the tutor in order to optimize performance. Authors should test for such a scenario in their current data.

As is articulated in our response to Essential revisions 1, we have now tested two possible 'calibration effects' and found no evidence for such phenomenon in our study. This suggests that the 'calibration effect' described previously in the work of Podos is supported by different mechanisms than the gene-experience mediated influences on learning that we describe here.

Finally, I had a few issues with the statistics: first, in the operant training group authors performed more than one statistical test, but did not account for multiplicity, which pose a risk of false discovery. Further, in one of those tests (Figure 4B), authors report one-tailed t-test, which seems unjustified (one-tailed should only be used in cases where it is impossible to obtain the other tail, which is not the case here if I understand the measures correctly). Given that the one-tailed test gave p<0.02, and that multiple comparisons were not accounted for, the correct p-value should be 0.08.

See response to Essential revisions 2, above.

Reviewer #3:This fascinating manuscript addresses the interaction of genetic predisposition and experience in determining adult behavioral phenotypes. However, in my view, needed data are lacking to justify the strong claims made about genetics in driving behavioral outcomes.For example, in the absence of a pedigree chart it is difficult to assess the apparent father-to-son heritability of a genetic predisposition to learn songs of a specific temporal structure. I am puzzled by the absence of any consideration of a maternal contribution to a genetic predisposition to song imitation – perhaps I have misunderstood, but this seems like an issue that the authors should address or acknowledge. A pedigree chart could certainly help sort this out.

Please see response to Essential revisions 3.

The role of genetic factors would also be strengthened by a multi-generational experiment showing the persistence of the genetic predisposition. While the manuscript's single-generation experiments are certainly compelling and clearly sufficient to demonstrate learning, a defining characteristic of a genetic influence is persistence across generations.

We appreciate that the reviewers, in consultation, decided that our ‘common garden’ approach was appropriate to support our conclusions. We provide below some additional discussion of the rationale for our experimental design.

Parent-offspring regression (as was used in our previous work (Mets and Brainard, PNAS, 2018) to assess heritability of song tempo) and other forms of single or double generation studies (such as twin studies or sibling-sibling comparisons) remain the most common methods for assessing heritability. As reviewer 3 points out, it is now also common to use a different approach based on a full pedigree. This approach, termed the ‘animal model’, uses a linear mixed-effects model and is especially useful in natural settings where environmental variables cannot be controlled experimentally. The estimates of heritability provided by the animal model are fundamentally derived from the same source as parent offspring regression, sib-sib correlation, or twin studies: the correlation between the genetic relatedness of individuals and the phenotype of interest. Pragmatically, there are two main reasons to use the animal model in place of a direct sibling-sibling or parent-offspring comparison. First, the animal model takes into account the relatedness of all individuals in a population, giving better statistical power than can be obtained in a similar population using only parent-offspring regression. Second, this approach allows statistical control over measured parameters (e.g. sex, water availability, food availability, mean temperature, etc.…) when these are not under experimental control. These two features of the animal model are particularly useful in wild populations where experimental control is impossible and the number of animals in a study may be limited. However, this approach has a limitation which is particularly problematic in the context of culturally learned behaviors generally, and song learning in particular. For phenotypes that are influenced by the phenotype of the parent through experience (trans-generational environmental effect) and where environmental influences are not experimentally controlled across the population, heritability can be over-estimated; an individual may behave in a way that is related to his cousin not because of the genetic similarity between those individuals, but because they have the same grandfather whose behavior was passed on to his offspring (and then their offspring) through cultural transmission. Therefore, without the ‘common garden’ approach used here, estimates of heritability for song phenotypes would be exaggerated.

Furthermore, the mixed-effects framework only allows statistical control of known and measured environmental variables whereas the 'common garden' approach controls for all environmental variables both known and unknown. Finally, the linear-mixed effects framework used in the ‘animal’ model only allows statistical control of environmental variables that exert a linear, additive influence on the phenotype of interest. We know from our previous work that some relevant environmental parameters have non-linear influences on song phenotypes (Mets and Brainard, 2018). We therefore felt the application of the animal model to our large pedigree could erroneously over-estimate the genetic influences on song and, instead, designed our experiments to directly control experiential factors.

Because I am enthusiastic about the research direction described in the manuscript, I would be happy to be corrected if the authors believe that I am in error. Comments regarding specific sections of the manuscript are listed below."In agreement with extensive prior research, we found that there was a broad range in the quality of song learning across individuals in our genetically heterogeneous Bengalese finch colony."• Could the authors provide references for "extensive prior research" on individual variation in song learning? A number of references are cited in the preceding paragraph, but these papers refer primarily to species or strain differences.

We agree that there is a relatively small amount of research specifically addressing variation in the quality of song learning under these conditions and, since this is not a major point, we have simply removed the clause 'In agreement with extensive prior research'.

• Could the authors define what they mean by "genetically heterogenous?" Are there breeding records that establish the degree of genetic heterogeneity within the colony? This seems important, given the role that genetic predisposition is argued to play in the analysis of the findings.

We now provide a pedigree for the birds included in the cross-fostering experiments (Figure 2—figure supplement 1). While this does not include the entire colony, the pedigree illustrates the general nature of the genetic diversity in our colony. Please see Essential revisions 3 for further details.

"Prior work indicates that the broad variation in the quality of song learning (Figure 1C) could be influenced by both experiential and genetic factors."• Seems like there are additional references that could be included here on the role of experience.

We have added several additional references.

"We compared learning for birds that were home-reared and tutored by their genetic fathers, to learning for birds that were cross-fostered and tutored by genetically unrelated adults."• What does "genetically unrelated" mean here? Were steps taken to insure a specific genetic distance between the adult pairs used in the home-reared and cross-fostered conditions?

We have now provided a pedigree to clarify the relational structure of individuals in this experiment (Figure 2—figure supplement 1). We have also changed the wording throughout to remove the use of ‘unrelated’, and have simply noted that we are comparing ‘home-reared’ and ‘cross- fostered’ animals. Please see Essential revisions 3 for further details.

Figure 2C: The effect shown in Figure 2C appears to be driven primarily by 3 of the 8 nests.• Could the authors justify or explain use of a signed-rank test to determine significance, as well as use of median SD scores?

There are several uncontrolled variables across nests that could potentially influence the measured quality and range of learning in a nest-specific fashion. These include, for example, the specific acoustic structure of the tutor song and how often it was produced. Such variation is likely to contribute to nest-specific differences in the magnitude or range of learning quality (SD scores). We therefore designed our experiments to enable paired comparisons between cross-fostered and home-reared birds within nests (in order to control for any nest-specific effects) and felt it appropriate to test the sign rather than the magnitude of differences between groups. We used median values to characterize learning within groups because the distribution of SD scores was γ, rather than Gaussian. We have confirmed that the significance of the results are unaltered if the mean is used instead of the median and/or if other statistical tests are used instead of a sign-rank test. These points are now noted in the second paragraph of the Results section and in the subsection “Statistics”.

• Throughout the manuscript, reporting the results of statistical tests is reduced to the name of the test and a p-value. Not sure if this is journal preference but seems non-standard.

We have expanded the Materials and methods section on statistics to provide further detail. Please see Essential revisions 2.

"Despite exposure to identical computer tutoring experience"• In the Materials and methods subsection “Computer tutoring”, the authors describe their perch-activated tutoring setup. Because tutor-song playback was contingent on the tutee hopping on the perch, one wonders if all birds were equally active in terms of their perch hopping. Although the number of tutor song playbacks per day was limited to 30, were there days when some birds received fewer because of less perch hopping? It would be helpful if the authors provided some evidence to support the statement that all birds received identical computer tutoring experience.

We have now analyzed the perch hopping logs for each bird to provide a more quantitative assessment of the amount of tutoring that each bird received and to examine any potential effects on learning. We confirmed that all birds received a similar amount of tutoring, and that we could not detect any relationship between variation in amount of tutoring and variation in the quality of learning. This is summarized in the following addition to the description of computer tutoring: “It sometimes took birds a few days to begin consistently actuating song playbacks. Nevertheless, all birds elicited at least 90% of the available song playbacks during the tutoring period (median = 96%), and there was no relationship between the number of playbacks that an individual heard and the quality of song learning.”

"Thus, differences in parental behavior and in individual experience with the live tutor could not account for observed variation in the quality of learning."• Seems like a nest-level analysis would be useful here. That is, the birds were raised to 35-40 days post-hatch by female foster parents. The text of the manuscript seems to assume that rearing by all female foster parents was of equivalent quality. A nest-level analysis of learning outcomes would address this assumption.

We have expanded the Materials and methods section covering female fostering to address this point. Specifically, we have added: “While the female foster mothers do not sing, they produce unlearned calls and influence other aspects of a hatchling’s experience, including feeding and social interactions, that could plausibly influence aspects of hatchling development and the quality of subsequent computer driven song learning. […] In order to prevent any ‘batch effects’, only a single egg/hatchling was raised at a time in any given foster nest.”

Figure 3B: Y-axis.• Perhaps indicate "Mean Song Divergence (bits)" on the y-axis, since median is used in Figure 2C?

The previous figure and legend were confusing with respect to which SD scores reflected individual bird values versus group values. We have now added labels and legend to clarify that the SD values in 3B and 3C are values for individual birds, and that the summary values in 3D indicate both medians (gray bars) and means (red points).

Figure 3B reports both the mean (gray bars) and the median (red dots) SD for each of the three groups depicted. We have therefore left the label unchanged. We have modified the legend to clarify